# Sodium Alginate/Chitosan Scaffolds for Cardiac Tissue Engineering: The Influence of Its Three-Dimensional Material Preparation and the Use of Gold Nanoparticles

**DOI:** 10.3390/polym14163233

**Published:** 2022-08-09

**Authors:** Nohra E. Beltran-Vargas, Eduardo Peña-Mercado, Concepción Sánchez-Gómez, Mario Garcia-Lorenzana, Juan-Carlos Ruiz, Izlia Arroyo-Maya, Sara Huerta-Yepez, José Campos-Terán

**Affiliations:** 1Process and Technology Department, Division of Natural Science and Engineering, Universidad Autonoma Metropolitana-Cuajimalpa, Ciudad de Mexico C.P. 05300, Mexico; 2Research Laboratory of Developmental Biology and Experimental Teratogenesis, Children’s Hospital of Mexico Federico Gomez, Ciudad de Mexico C.P. 06720, Mexico; 3Department of Reproduction Biology, Division of Biological and Health Sciences, Universidad Autonoma Metropolitana-Iztapalapa, Ciudad de Mexico C.P. 09340, Mexico; 4Research Laboratory of Hematooncology, Children’s Hospital of Mexico Federico Gomez, Ciudad de Mexico C.P. 06720, Mexico

**Keywords:** alginate, chitosan, scaffolds, nanoparticles, cardiac tissue engineering

## Abstract

Natural biopolymer scaffolds and conductive nanomaterials have been widely used in cardiac tissue engineering; however, there are still challenges in the scaffold fabrication, which include enhancing nutrient delivery, biocompatibility and properties that favor the growth, maturation and functionality of the generated tissue for therapeutic application. In the present work, different scaffolds prepared with sodium alginate and chitosan (alginate/chitosan) were fabricated with and without the addition of metal nanoparticles and how their fabrication affects cardiomyocyte growth was evaluated. The scaffolds (hydrogels) were dried by freeze drying using calcium gluconate as a crosslinking agent, and two types of metal nanoparticles were incorporated, gold (AuNp) and gold plus sodium alginate (AuNp+Alg). A physicochemical characterization of the scaffolds was carried out by swelling, degradation, permeability and infrared spectroscopy studies. The results show that the scaffolds obtained were highly porous (>90%) and hydrophilic, with swelling percentages of around 3000% and permeability of the order of 1 × 10^−8^ m^2^. In addition, the scaffolds proposed favored adhesion and spheroid formation, with cardiac markers expression such as tropomyosin, troponin I and cardiac myosin. The incorporation of AuNp+Alg increased cardiac protein expression and cell proliferation, thus demonstrating their potential use in cardiac tissue engineering.

## 1. Introduction

Acute myocardial infarction (AMI) remains the primary cause of death worldwide. AMI occurs when blood flow to the coronary arteries is blocked, leading to necrosis, tissue remodeling and fibrosis that can cause progressive cardiac damage and heart failure [1]. A major problem in the recovery of AMI patients is the low proliferation percentage for heart cell regeneration [2]. For effective AMI treatment, it is necessary to prevent tissue remodeling, attenuate scar formation and promote cell proliferation and regeneration to replace damaged tissue. For this reason, tissue engineering emerges as a therapeutic alternative for the development of functional tissue that can be used for the regeneration of affected tissues. In cardiac tissue engineering, attempts are made to combine cells with biocompatible materials to generate a three-dimensional construct that can restore the damaged myocardium [3]. The ideal scaffolds for cardiac repair must have high porosity and biocompatibility as well as be permeable to nutrients and metabolic waste. In addition, they must have an adjustable degradation time to minimize the formation of fibrous capsules and promote incorporation into the host tissue to avoid a chronic inflammatory response [4]. Likewise, they must recreate the microenvironment, structure and three-dimensional organization of the myocardium; improve cell survival and promote cell adhesion, differentiation and maturation. In addition, the scaffolds must allow for vascularization to ensure the flow of oxygen and nutrients to the cells and favor the transmission of electrical and mechanical impulses for proper host-tissue coupling [5]. 

Natural polysaccharides such as alginate and chitosan have been widely used for tissue engineering because of their biocompatibility, biodegradability and structural similarity to the extracellular matrix components [6,7,8,9,10]. Chitosan, a polycationic polymer (the presence of positively charged amine groups), promotes the cell adhesion, proliferation and differentiation of different cell types [7,11]. Alginate, a polyanionic polymer (presence of negatively charged carboxyl groups), promotes regeneration and favors the vascularization and restoration of electrical conductivity and cell growth [6]. The chemical natures of alginate and chitosan polymers make them sensitive to pH (pKa = 3.4–3.7 and pKa = 6.3, respectively) of aqueous media, swelling in opposite directions [5].

Previous reports have shown that alginate/chitosan scaffolds improve mechanical and biological properties [8,9,12], in addition to promoting growth and maintaining cardiac cell viability [10,13,14,15,16]. They have a gradual degradation and favor cell retention, survival and migration to the affected area, allowing the formation of blood vessels, with reduction in fibrosis and hypertrophy area [14,17]. 

The addition or functionalization of biomaterials with metallic nanomaterials can improve the physical and electrical properties of the scaffolds [18,19,20,21]. Recently, the need and importance of designing and developing new cardiac patches based on conductive biomaterials for possible therapeutic application has been reported [22,23,24]. The properties of gold nanoparticles (AuNp) promote cardiac cell growth and contractility [25,26,27,28]. Biomaterials functionalized with AuNp favor cardiomyocyte elongation and alignment, with an increased expression of cardiac proteins and improved cell contraction [29,30,31,32,33]; however, the use of AuNp has not been studied in alginate/chitosan scaffolds. 

Although there have been good results in the use of alginate/chitosan scaffolds for cardiomyocyte growth, it is interesting to test the use of star-type AuNp and tubular AuNp+Alg in this type of scaffold to favor cell adhesion and growth and increase cardiac protein expression, identifying the best conditions for scaffold fabrication for therapeutic purposes. Thus, the aim of this work was to compare different methods of fabrication of alginate/chitosan scaffolds to improve swelling percentages, permeability, porosity and degradation rate and to evaluate the effect of the functionalization of the proposed scaffolds with AuNp and AuNp+Alg on cardiomyocyte growth.

## 2. Materials and Methods

### 2.1. Preparation of Alginate–Chitosan Scaffolds

Sodium alginate (Sigma Aldrich, Mannheim, Germany, # 9005-38-3) and chitosan (medium molecular weight, Sigma Aldrich, Mannheim, Germany, # 448877) (0.75–1.25% *w*/*v*) powders were mixed and dissolved in ultrapure water (Milli-Q system, 18.2 M-cm) and acetic acid (1% *w*/*v*, Sigma Aldrich, Mannheim, Germany, #1005706). The pH was adjusted to be between 5 and 6 to favor the interaction between the biomaterials. The resulting mixture/solution was ready for undergoing the corresponding experimental method (1 to 4), see Figure 1. In all cases, the scaffold solutions placed within the 24-well box were frozen at −20 °C for 12 h and subsequently freeze-dried in a lyophilizer (Labconco Corporation, Kansas City, MO, USA) for 8 h at −49 °C under vacuum with a pressure of 0.100 mBar.

Method 1 (without sonication). An alginate–chitosan solution of 0.5 mL was deposited in each well of a 24-well box. Then it was frozen, lyophilized and crosslinked with 1 mL of 1% *w*/*v* calcium gluconate in water for 15 min. Washes were performed with ultrapure water and again frozen at −20 °C for 12 h. Finally, they were lyophilized again for 8 h.

Method 2 (with sonication). The alginate–chitosan solution was sonicated (Sonics Vibra Cell VCX 750, Newtown, CT, USA) for 5 min at 20 kHz and 750 W. Subsequently, the alginate–chitosan solution was deposited in each well of a 24-well box. After freezing and lyophilization, crosslinking was performed with 1% calcium gluconate for 15 min. Subsequently, they were washed with ultrapure water, frozen and lyophilized for 8 h.

Method 3 (longer crosslinking time). The alginate–chitosan solution was deposited in each well of a 24-well box. After freezing and lyophilization, crosslinking was performed with 1% calcium gluconate for 30 min. Subsequently, washes were performed with ultrapure water, frozen and freeze-dried for 8 h.

Method 4 (with sonication and longer crosslinking time). The alginate–chitosan solution was sonicated at 20 kHz and 750 W. The solution was then deposited in the 24-well box, frozen and lyophilized. Crosslinking was performed for 30 min with 1% calcium gluconate. Subsequently, washes were performed with ultrapure water, frozen and freeze-dried for 8 h.

### 2.2. Synthesis of Metallic Nanoparticles 

For the functionalization of alginate–chitosan scaffolds, metallic nanoparticles (Np) were prepared using a novel methodology with modifiable topography. This synthesis method is based on the preparation of citrate-stabilized gold nanoparticles [34], which are attached to the surface of a polymeric core consisting of Poly(D,L-lactide-co-glycolide) acid (PLGA) and stabilized with the copolymer Pluronic F-127. After the preparation of these metallic nanoprecursors, the growth of a gold shell is promoted on their surfaces, finally generating gold nanoparticles (AuNp). 

The AuNp synthesis process is described below: first, the synthesis of PLGA (Sigma Aldrich, 102229183, Mannheim, Germany) cores was carried out. For this, a 10% *w*/*v* solution of PLGA in acetone (C_3_H_6_O) was prepared (Sigma Aldrich, #67641, Mannheim, Germany), which was drip-added to 38 mL of a 1% *w*/*v* aqueous solution of the triblock copolymer Pluronic F-127 (C_3_H_6_O-C_2_H_4_O)x (Sigma Aldrich, #9003116, Mannheim, Germany), under stirring (250 rpm) at a constant temperature of 10 °C. Subsequently, it was homogenized for 10 min in an ice bath with a sonic tip of 750 W, 20 kHz frequency and 40% amplitude and left in agitation for 4 h. At this stage of the AuNp synthesis, a modification to the PLGA nuclei preparation method was also carried out, which consisted of coating some of them with a sodium alginate solution. Briefly, 20 mL of PLGA nuclei was mixed with 1 mL of 1% (*w*/*v*) sodium alginate and the mixture was stirred (250 rpm) for 4 h at room temperature. PLGA nuclei were purified through three cycles of centrifugation (9000 rpm, 18 °C) and resuspension. The supernatant was discarded, and the pellets were dispersed in 30 mL of ultrapurified water. Subsequent to the PLGA cores preparation, gold nanoseeds were synthesized. These were obtained by mixing the following solutions: 0.125 mL of 0.01 M chloroauric acid (HAuCl_4_) (Sigma Aldrich, 1001642619, Mannheim, Germany), 10 mL of 0.0256 M trisodium citrate (Na_3_C_6_H_5_O_7_) (Sigma Aldrich, 1001851140, Mannheim, Germany) and 0.3 mL of 0.1 M sodium borohydride (NaBH_4_, Sigma Aldrich, 1002918750, Mannheim, Germany). The latter was added at a temperature of 4 °C. The formation of gold nanoprecursors was carried out by mixing both the PLGA cores and alginate-modified PLGA cores (PLGA+Alg) with the gold nanoseeds in a 1:1 ratio, under constant stirring at 250 rpm for 24 h, followed by centrifugation (7000 rpm, 18 °C) for 20 min. The supernatant was discarded and the pellets were dispersed in 30 mL of ultrapurified water. The gold nanoprecursors were sonicated for 10 min to avoid particle aggregation. Once the PLGA cores were coated with the gold nanoseeds, the metal shell was assembled on their surface by mixing 2.025 mL of AuNp nanoprecursors with 45 mL of a solution containing 3.69 mM potassium carbonate (K_2_CO_3_, Sigma Aldrich, 1002055627, Mannheim, Germany) and 0.025 M gold (III) chloride trihydrate (HAuCl_4_-3H_2_O) (Sigma Aldrich, 1001642619, Mannheim, Germany). The latter was carried out under stirring (250 rpm) at room temperature. After 5 min, 225 μL of fresh 0.5 M ascorbic acid was added to the mixture. 

The above was performed for both PLGA cores and sodium-alginate-modified cores (PLGA+Alg). The functionalization of the alginate–chitosan scaffolds was carried out through the addition of AuNp in a calculated concentration range between 1 × 10^−12^ and 3 × 10^−9^ mg/mL. The AuNp concentration was calculated using the Lambert–Beer equation. For the above, the determination of the molar extinction coefficient was calculated based on the following equation:(1)$$\varepsilon =\frac{N_{a}\cdot \sigma }{2303}$$
where *ε* is the molar extinction coefficient, *N_a_* is Avogadro’s number, *σ* is the effective cross section in cm^2^ and 2303 is (ln 10) × 1000.

### 2.3. Characterization of Alginate/Chitosan Scaffolds

The alginate/chitosan scaffolds synthesized by the four methods and their subsequent modification with AuNp or AuNp+Alg were characterized by means of swelling, permeability, porosity and degradation tests in different aqueous media, recording their weight and thickness prior to their use. Fourier transform infrared spectroscopy (FTIR) was used as a chemical characterization technique.

### 2.4. Swelling Degree Studies 

For swelling measurements over time in a typical experiment, the scaffold (approximately 1.3 cm in diameter) is placed in contact with 2 mL of the aqueous medium at a constant temperature of 20 °C at different times, after which the sample is removed from the medium, weighed and placed back in the medium. The degree of swelling, *S*, in %, was calculated gravimetrically using Equation (2), where *W*_s_ and *W*_0_ are the weights of the swollen scaffold and dry scaffold (initial), respectively. The following aqueous media were used: ultrapure water (pH 7), PBS (phosphate buffer saline, pH 7.4) and buffers prepared with mixtures of Na_2_HPO_4_/citric acid solutions to obtain pHs of 3, 5, 8 and 9 (measured with the Conductronic potentiometer model PC45 (Puebla, Mexico).
(2)$$\mathrm{Swelling},\;S,\;\left(\%\right)=\left(\frac{W_{S}-W_{0}}{W_{0}}\right)\times 100$$

### 2.5. Permeability Value

The intrinsic permeability coefficient (*k*) was calculated according to Darcy’s law:(3)$$k=K\frac{\mu }{\rho g}$$
(4)$$K=\frac{a}{A}\frac{L}{t}\ln\frac{H_{1}}{H_{2}}$$
where (*μ*) is the viscosity of the medium, (*ρ*) is the density of the medium, (*g*) is the gravity acceleration, (*a*) is the tube area, (*A*) is the cross-sectional area at the sample flow, (*L*) is the sample thickness (in this case of the scaffold) and (*H*_1_) and (*H*_2_) are the initial and final heights of the tube through which the medium passes. We used 27.5 cm for *H*_2_. A detailed description of the custom device is presented in the Appendix A, which is similar to the other reported systems [35,36]. 

### 2.6. Porosity

The scaffolds’ porosity was determined by the liquid displacement method and ethanol was used as the penetrating medium because it does not induce shrinkage or swelling, is not a solvent for polymers and is able to easily penetrate the pores. Each scaffold was placed in a cylinder with a known volume of ethanol, in which it was left for 48 h, the scaffold was removed, and the final volume was recorded. Finally, the following equation was used:(5)$$\mathrm{Porosity}\;\left(\%\right)=\left(\frac{W_{S}-W_{0}}{\rho V}\right)\times 100$$
where (*W*_S_) is the weight of the saturated scaffold, (*W*_0_) is the initial weight of the scaffold, (*ρ*) is the ethanol density and (*V*) is the volume of liquid displaced. 

### 2.7. Degradation

Degradation studies were divided into two main studies. The first study was performed at conditions similar to cell cultures in which the scaffolds (5 mm in diameter approximately) were immersed in M199 medium (11150067, Gibco, Thermo Fisher Scientific, Waltham, MA, USA) supplemented with fetal bovine serum (A4766801, Gibco, Thermo Fisher Scientific, Waltham, MA, USA) to a pH of 7.4 and were placed in an incubator at 37 °C for 17 days to evaluate their degradation degrees at the following times: days 1, 2, 3, 6, 10 and 17. At all times, as much water as possible was removed from the container with the scaffold and weighed, and new medium was placed before returning to the incubator. For the second study, the material degradation (approximately 1.3 cm in diameter) exposed to different aqueous media for 7 days at laboratory conditions (20 °C) was calculated. In a typical experiment, for the scaffold immersed in the medium, 2 mL was used with each wash and proceeded with the following times and number of washes: twice for 15 min, once for 16 h and twice for 15 min. Finally, as much water as possible was removed from the container with the scaffold, frozen at −20 °C for at least 12 h and freeze-dried in a lyophilizer at −49 °C and a pressure of 0.09 mBar for 6 h (followed gravimetrically until there was no weight change). The degradation degree, *D*, in %, was calculated according to Equation (5), where *W*_F_ and *W*_0_ are the final weight of the dry scaffold and exposed to either degradation condition and the weight of the initial dry scaffold (before being exposed to either degradation condition), respectively.
(6)$$\mathrm{Degradation}\;\mathrm{degree}=n,\;D,\;\left(\%\right)=\left(\frac{W_{0}-W_{F}}{W_{0}}\right)\times 100$$

### 2.8. Infrared

Fourier transform infrared spectra with attenuated total reflectance (FTIR-ATR) were taken from 650 cm^−1^ to 4000 cm^−1^ using a Perkin–Elmer model 100 spectrometer (Waltham, MA, USA) equipped with a diamond tip.

### 2.9. Characterization of the Metallic Nanoparticles

The particle sizes, their distributions and the zeta potentials of AuNp and AuNp+Alg were analyzed by dynamic light scattering using a Nanosizer Nano ZS (Malvern Instruments Ltd., Malvern, UK). Samples were diluted 1:100 in ultrapure water and analyzed at 25 °C, with a scattering angle of 90. 

The Np morphology was determined by scanning electron microscopy, using a TM3030PLUS scanning electron microscope (Hitachi, Germany) with an operating voltage of 15 kV; and moreover, the AuNp morphology was examined using a JEM-1010 transmission electron microscope (Jeol, Tokyo, Japan) with a voltage of 60 kV. The AuNp aqueous dispersion was diluted 1:100 and 15 µL was deposited on a copper grid (200 mesh). The grid was allowed to dry at room temperature before analysis. UV-Vis spectroscopy analysis was conducted in order to visualize the plasmon of the gold nanoparticles. 

### 2.10. Primary Culture of Chicken Embryonic Cardiomyocytes

Cardiomyocytes were obtained from chicken embryos after 7 days of incubation. Detailed information on cardiomyocytes isolation and culture is presented in the Appendix A. For each scaffold, 1 × 10^6^ cells were cultured and incubated for 7 days at 37 °C in supplemented medium 199. Animal use protocols and study procedures were based on the Official Mexican Standard (NOM-062-ZOO-1999). The project was approved by the research, ethics and biosafety committees of the Hospital Infantil de México Federico Gómez (HIM-2020-059).

### 2.11. Indirect Cytotoxicity Assay

To demonstrate that the different elaborated scaffolds did not generate cytotoxic particles, an indirect cytotoxicity assay was performed using the MTT method. The viability of cells cultured with medium that had contact with scaffolds functionalized with AuNp and AuNp+Alg was also compared. Indirect cytotoxicity assays were performed using a monolayer of cardiomyocytes by triplicate. 

### 2.12. Scanning Electron Microscopy

Scaffolds and constructs were fixed with glutaraldehyde (4%) and dehydrated through a series of graded ethanol concentrations (50° to absolute). They were critical-point-dried (Samdri 789A, Tousimins Research Co., Rockville, MD, USA) and coated with gold film (Denton Vacuum Desk 1A, Cherry Hill Industrial Center, Moorestown, NJ, USA). The samples were observed under a JEOL JSM 5300 (Tokyo, Japan) scanning electron microscope, and the accelerating voltage was 15 kV.

### 2.13. Histological Analysis

Constructs were fixed with paraformaldehyde (4%) and processed according to standard histological technique. Transverse sections of 3 µm thickness were made and stained with hematoxylin–eosin (H&E) and scanned and digitized with Aperio CS2 equipment (Leica Biosystems, Deer Park, IL, USA).

### 2.14. Immunohistochemical Analysis

The sections were deparaffinized and subjected to antigenic recovery in sodium citrate buffer (pH 6). Endogenous peroxidase blocking was performed with hydrogen peroxide (3%) for 30 min. A nonspecific binding blockade was performed for 3 h. The sections were incubated with the primary antibodies anti-tropomyosin (SC:74480, 1:500) and anti-PCNA (AB-2426, 1:1000) at room temperature overnight. They were then incubated with horseradish peroxidase (HRP)-conjugated secondary antibody for 30 min. The antigen–antibody complex was revealed with an immunodetection kit (Vector Laboratories, Inc. Cat # 30026, Berlingame, CA, USA) and counterstaining was performed with hematoxylin. Quantitative analysis (Intensity (Int)) was performed by digital pathology.

### 2.15. Western Blot

Total protein extraction from the constructs was performed with lysis buffer (T-PER, Thermo Fisher Scientific, Waltham, MA, USA) and added to protease inhibitor (Sigma Aldrich, Mannheim, Germany). Protein concentration was quantified using the direct microdrop method (NanoDrop lite, Thermo Fisher Scientific, Waltham, MA, USA). A total of 30 µg of total protein was subjected to 10% sodium dodecylsulfate polyacrylamide gel electrophoresis (SDS-PAGE) and transferred to a nitrocellulose membrane (Bio-Rad). Antibodies were used at the following dilutions: primary antibodies troponin I (SC-365446 1:1000), MYH (SC-376157, 1:1000) and GAPDH (SC-48167), followed by incubation with HRP-coupled secondary antibodies in blocking solution for 1 h at room temperature, anti-mouse (SC-516102, 1:10,000) and anti-goat (SC-2020, 1:20,000). Finally, immunodetection by chemiluminescence (Super Signal^®^ West Femto, Thermo Scientific, Waltham, MA, USA) was performed with autoradiographic plates. Densitometry analysis was performed with Image J software, version 1.45 (National Institute of Health, Bethesda, MD, USA).

### 2.16. Statistical analysis

GraphPad Prism 9 software (San Diego, CA, USA) was used to perform the statistical analysis, using a significance value of *p* < 0.001. Data are presented as the mean ± standard error (SE). An ANOVA was performed to evaluate physical changes in scaffold fabrication and changes in percent swelling and permeability between scaffold fabrication methods and to analyze cell viability. To compare the four scaffold fabrication methods with respect to functionalization (without Np, with AuNp and with AuNp+Alg) a multivariate analysis was performed, with Tukey’s post hoc test for multiple comparisons.

## 3. Results

### 3.1. Characterization of Alginate–Chitosan Scaffolds

Method 1 (without sonication) had the highest weight with respect to the other three fabrication methods (*p* < 0.001). Regarding thickness, the scaffolds generated by Method 2 (with sonication) were the thinnest, presenting significant changes with respect to Methods 1 and 3 (*p* < 0.001), see Appendix A.

### 3.2. Swelling Degree 

In Figure 2A, the swelling percentages of the scaffolds according to the processing method can be observed. The swelling percentage of the scaffolds was higher than 2500% in three of the methods. The scaffolds elaborated with Method 3 (30 min crosslinking) were the ones with the highest swelling percentage. Figure 2B shows that the swelling percentage increases when functionalizing the scaffolds with AuNp and AuNp+Alg, particularly in Method 3, which had the highest values.

The scaffolds obtained with Method 3 were studied in their swelling in different aqueous media (pH dependence) over 7 days (Figure 3). Similar behavior can be observed in each medium over time, in which the swelling increases, reaches a maximum and does not decrease significantly at the total time studied. The Table 1 shows the percentage reached with respect to the maximum swelling (*S*_max_) of the scaffolds. At 5 min, the hydrogels reached between 53 and 94% of the *S*_max_, and at 60 min, most of the scaffolds reached between 90 and 100% of the *S*_max_. The swelling dependence of the alginate/chitosan scaffolds on pH showed a decrease in the swelling when going from acidic to basic pHs, from 2945 to 1880% (Figure 3C). 

### 3.3. Permeability 

Method 2 (sonication) has the highest permeability coefficient, being more than twice the one obtained with the other scaffold processing methods (Figure 2C). When functionalizing the scaffolds with AuNp and AuNp+Alg, a slight increase in permeability is observed in the four proposed scaffold elaboration methods (Figure 2D).

### 3.4. Porosity 

It is observed that the scaffolds of Method 3 (30 min crosslinking) are the most porous, and this property increases to more than 95% when functionalized with AuNp (Figure 2E).

### 3.5. Degradation 

Degradation analysis under cell culture conditions was performed on scaffolds made with the four proposed methods (Figure 2F) and on scaffolds of the four methods functionalized with AuNp (Figure 2G). It is observed that the scaffolds of Method 1 (without sonication) are the ones that degrade slower without AuNp, but when functionalized with AuNp, they degrade faster. In the case of Method 4, the opposite happens; they are the scaffolds that degrade faster without AuNp and with AuNp they degrade slower. In general, they degrade slightly faster with AuNp, with the exception of Method 3 (longer crosslinking time) where functionalization with AuNp causes them to degrade slightly slower, although the changes are not statistically significant.

The scaffolds’ degradation by Method 3 (more crosslinking time) with respect to time and subjected to different aqueous media for 7 days are shown in Figure 3D. For all samples, there is a weight decrease, corroborating a degradation between 8 and 35%, and, as observed, it increases with increasing pH.

### 3.6. FTIR-ATR

Figure 4 shows the chemical characterization (IR spectra) of the alginate/chitosan scaffolds before and after crosslinking by Method 3 (30 min crosslinking), followed by the incorporation of AuNp and AuNp+Alg, in addition to the initial sodium alginate and chitosan spectra. For sodium alginate (Figure 4(A1)), the presence of the characteristic groups is confirmed [37]. Figure 4(A2) shows the characteristic peaks of chitosan [38]. In the case of the scaffold with alginate/chitosan before crosslinking (Figure 4(A3)), peaks that overlap due to the presence of both components can be observed (follow the dotted lines). The spectrum of the scaffold crosslinked with calcium gluconate by Method 3 (Figure 4(A4)) is similar to the spectrum of the scaffold before crosslinking, but an incorporation of OH and COO– due to the gluconate is present, and an increase in the intensity of the –CH peak is observed. The IR spectra of the scaffolds crosslinked and with the incorporation of AuNp and AuNp+Alg are presented in Figure 4(A5,A6), respectively; they are similar to the scaffolds before doping with AuNp, see Appendix A.

### 3.7. Characterization of AuNp

The obtained metallic nanoparticles presented an average hydrodynamic diameter of 74.5 ± 1nm and 91.1 ± 1 nm for PLGA and PLGA+Alg core-shell nanoparticles, respectively. As for the surface charge value, it was determined to be −25.4 ± 0.8 mV for PLGA-cored samples and −36.8 ± 1 mV for those modified with sodium alginate (see Appendix A). 

Particle morphology was determined by SEM and transmission electron microscopy (TEM). According to these techniques, AuNp presented a spheroidal structure (Figure 4B), while AuNp+Alg showed cylindrical particle characteristics (Figure 4C). The UV-Vis spectra of AuNp and AuNp+Alg solutions (before dilution) were taken (Figure 4D), presenting both a plasmon starting at ~500 nm and reaching a maximum wavelength at 571 nm and 639 nm, respectively, confirming the sizes observed by TEM. As a reference, Peña et al. showed a maximum plasmon at a wavelength of 528 nm for homogeneous spherical AuNp ~32 nm of diameter [30].

### 3.8. Cytotoxicity

The presence of Np does not affect cell viability. The scaffolds elaborated by Methods 1, 2 and 3 were the ones that allowed the maintenance of up to 80% of viable cells (Figure 2H). The results demonstrate that the alginate/chitosan scaffolds elaborated by the different proposed methods without Np and functionalized with AuNp and with AuNp+Alg do not release toxic components that could negatively impact the viability of the cells cultured on them.

### 3.9. Scanning Electron Microscopy (SEM)

The electromicrographs of the scaffolds show a porous appearance (Figure 5). The presence of spheroids of different diameters can be observed (Figure 5B,G–I,L). Functionalization with Np allowed the increase in the number, size and distribution of spheroids, especially for the scaffolds of Method 3 (Figure 5H,I). 

### 3.10. Histological Analysis

Histological analysis shows that there was high cell migration and penetration through the pores of the scaffolds. Consistent with the SEM, the scaffolds of Method 3 were the ones that presented the largest spheroids, reaching 200 µm in diameter, in addition to being present both in the periphery and inside the scaffold (Figure 6). All the scaffolds with Np presented a greater number of spheroids compared to the scaffolds without Np (Figure 6B,E,H,K). However, spheroids were observed in all four proposed methods.

### 3.11. Immunodetection of Proliferation and Cardiomyocyte Markers

Immunohistochemical analysis was performed to measure the levels of PCNA (proliferation marker, Figure 7) and tropomyosin (Figure 8). There was increased proliferation in the scaffolds prepared by Methods 3 and 4, and in particular in Method 3, functionalization with AuNp+Alg significantly increased cell proliferation compared to scaffolds without Np (Figure 7M). For the case of tropomyosin, an increase in this protein was observed mainly in the periphery of the spheroid in the scaffolds elaborated by Methods 1, 3 and 4. The scaffolds elaborated by Methods 3 and 4 showed the highest tropomyosin expression, particularly when functionalization with AuNp+Alg was present. Figure 8C shows the interaction between the scaffold and the cardiomyocytes (amplification is shown in the Appendix A).

Immunodetection by western blot revealed troponin I and cardiac myosin expression in all scaffold elaboration methods (Figure 9A). In addition, densitometric analysis was performed to demonstrate that for myosin, in the case of the scaffolds of Methods 2 and 4, functionalization with both AuNp and AuNp+Alg caused an increase in expression. In the case of troponin I, it can be seen that in the scaffolds prepared by Methods 2 and 3, functionalization with AuNp+Alg also promoted increased expression (Figure 9B). 

## 4. Discussion

AMI generates a high incidence of deaths worldwide; thus, it is necessary to develop new therapeutic strategies to regenerate cardiac tissue that often loses its function [39]. 

In this work, four different methods for the elaboration of sodium alginate–chitosan scaffolds are proposed. Crosslinking between polymers is important because it provides the scaffolds with increased stability, higher mechanical strength and hydrolysis resistance [40]. In addition, crosslinking agents promote chemical interactions between exposed functional groups. Although various crosslinking agents have been proposed to improve the biological and mechanical properties of these hydrogels [13,41], in this work the use of calcium gluconate as a crosslinking agent was proposed, favoring cell adhesion and cardiomyocyte growth. In addition, it was found that the fabrication method has an important impact on the structure and properties of the scaffolds and that in general the proposed methods favor the cardiomyocytes growth and contractility. 

Scaffolds that have low electrical conductivity can be modified with gold, carbon, selenium or silver Np to increase their conductivity and improve their biocompatibility [29,42]. In particular, AuNp considerably increases scaffold conductivity, favors cell organization, enhances cell contractile activity and promotes cardiomyocyte maturation, proliferation and migration [21,27,31,33,43]. 

Nanoparticles tend to be more stable and interact better with biological systems when they are coated with materials of biological origin such as alginate [44,45] or albumin [46,47], among others. In this work, we proposed to use sodium alginate as a stabilizing agent and we evaluated the physicochemical and biological properties of the scaffolds and the metallic nanoparticles that were developed. 

It has been reported that a high swelling percentage is favorable for cell growth, adhesion and vascularization [48]. Our results show that the scaffolds that were prepared had swelling percentages above 3000%, a percentage higher or similar to what is reported in other alginate/chitosan scaffolds, in addition to corroborating their pH dependence [14,49]. Baei et al. [27] reported that the addition of AuNp in chitosan scaffolds decreases the hydrophilic functional groups exposed, which translates into lower swelling percentages. Contrary to what they report, in our alginate/chitosan scaffolds, when functionalized with AuNp, the swelling percentage increased, probably due to the fabrication and crosslinking method that was proposed.

Permeability is a property that is directly related to the pore interconnectivity degree, as it controls the nutrient flow through the scaffold to the cells migrating inside it, thus achieving efficient cell growth [50]. Tresoldi et al. [51] reported permeabilities of 9 × 10^−15^ m^2^ in alginate–gelatin scaffolds, while in studies performed by Rai et al. [52], permeabilities of 2 × 10^−15^ m^2^ were obtained for fibrous poly(glycerol sebacate)-poly(ε-caprolactone) scaffolds. In the case of this work, a 1 × 10^−8^ m^2^ permeability was obtained in all the proposed fabrication methods, which favors the flow of nutrients and oxygen to the interior of the scaffolds, for greater cell proliferation. 

It was demonstrated that our scaffolds were highly porous, which is a feature that directly influences cell penetration and nutrient and oxygen transport [53], with adequate pore size to favor cell infiltration and colonization. The scaffolds elaborated by Method 3 (longer crosslinking time) were the most porous, and functionalizing them with AuNp was found to increase their porosity, to increase their swelling percentage and to decrease their degradation time, which favors the formation of cardiac tissue.

The degradation rate depends on the regeneration rate of the tissue to be replaced. In this work, the degradation percentage was analyzed by placing the scaffolds in culture medium supplemented with fetal bovine serum, which had not been previously reported, to know their behavior under culture conditions. Similar to what has been reported in other works, functionalization with AuNp decreased the degradation rate [54], which in our case favored cardiomyocyte cell growth. 

The physicochemical characteristics of proposed gold Np (size and zeta potential) were similar to those reported with potential applications in the biomedical field, specifically in tissue engineering [18,19,20,21]. Stable Np do not agglomerate and help improve the electrical conductivity of the scaffold. Our AuNp+Alg had a Z potential (−36 mV) similar to the AuNp+Alg (−30 mV) elaborated by Shen, K, et al. [45], which were stable and helped to generate percentages of cell viability around 95%.

The difference in morphological characteristics between AuNp and AuNp+Alg could be explained by the presence of sodium alginate surrounding the PLGA nuclei, which acts as a structural growth director in a favored specific direction, as has also been reported by Pal et al. [55].

Nanoparticles used in tissue engineering have various sizes. Small Np (20 nm) [56] and large Np (156 nm) [19] have been reported to increase cell viability by up to 60%, suggesting that the size of the Np does not influence their biocompatibility. In our case, the AuNp+Alg (91 nm) were the ones that increased cardiac cell proliferation (Figure 6).

This work demonstrated that natural biomaterials such as alginate and chitosan are not cytotoxic and allow for cardiomyocyte growth as reported in other study groups [57]. The final architecture of our constructs allowed for the adhesion, growth and maintenance of cardiac cell integrity; the cultures remained healthy for 7 days; and the cells were able to enter the scaffolds, grow between the pores and form spheroids. In addition, histological and molecular analysis allowed us to observe the expression of the characteristic proteins of cardiac tissue, which increased with the incorporation of the AuNp+Alg that was proposed (Figure 7 and Figure 8), which is in agreement with the work of Dvir et al., where functionalized scaffolds were shown to increase the expression of cardiac markers [58]. In our case, it was probably due to the biological properties reported for alginate [6]. 

In addition, an increase in cardiac cell differentiation has been reported when aggregated into spheroids, particularly if these aggregates can bind to a substrate [59], a situation that occurred in this study system.

Similar proliferation was observed in all scaffolds, but the proposed new functionalization with AuNp+Alg considerably increased cell proliferation. 

It is interesting to note that in addition to functionalization, the proposed processing methods also have an impact on the expression of cardiac markers, confirming that scaffold topology plays a fundamental role in cell distribution and development. Several factors influence bioengineered tissue generation: on the one hand, the scaffolds and their structure, and on the other hand, the integrity of the cells to be cultured and the appropriate culture conditions. Taken together, the results suggest that these cardiac tissue constructs may be a novel implant material to study cardiac patch–cell interactions and can be used to study their biocompatibility, degradation and vascularization in an in vivo model.

## 5. Conclusions

The data as a whole demonstrate that the proposed alginate/chitosan scaffolds elaborated by the working group favor the growth of cardiomyocytes and spheroid formation and that functionalization with AuNp and AuNp+Alg not only increases cell proliferation but also promotes the increase in cardiac proteins such as troponin I, myosin and tropomyosin. 

The presence of a carbohydrate-based biopolymer such as sodium alginate in AuNp+Alg proved to be an effective means to modify the structural and physicochemical properties of AuNp. Furthermore, it was demonstrated that AuNp+Alg scaffolds can be used as a therapeutic alternative in cardiac tissue engineering.

The proposed scaffolds were found to be useful for cardiomyocyte growth, which presents the possibility of applying them to the generation of a patch that can be used as a therapeutic alternative in AMI.

## 6. Patents

Soporte de hidrogel de alginato y quitosano para crecimiento de tejidos. Beltran NE, Francisco E., Vaquero D, Arroyo I. Instituto Mexicano de la propiedad industrial, MX/a/2020/012621. 24/11/2020.

## Figures and Tables

**Figure 1 polymers-14-03233-f001:**
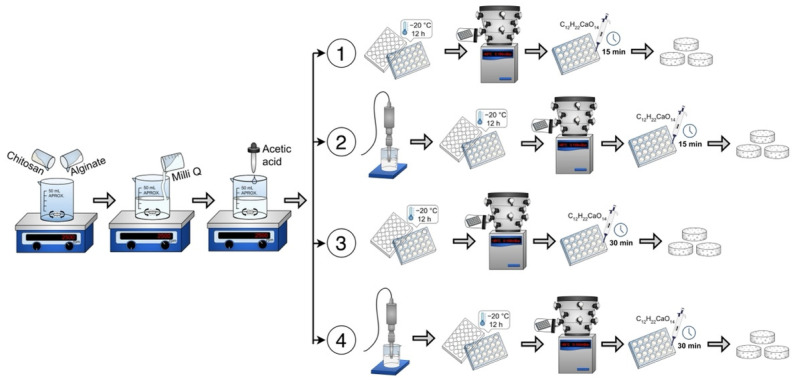
Four scaffold preparation methods: Method 1 (without sonication), Method 2 (with sonication), Method 3 (longer crosslinking time) and Method 4 (with sonication and longer crosslinking time).

**Figure 2 polymers-14-03233-f002:**
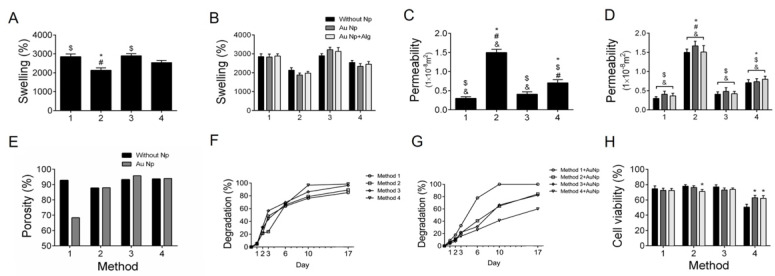
(**A**) Swelling percentages (%) for the four scaffold preparation methods. (**B**) Swelling percentages for the four preparation methods functionalized with AuNp and AuNp+Alg. (**C**) Permeability (m^2^) for the four scaffold preparation methods. (**D**) Permeability for the four preparation methods functionalized with AuNp and AuNp+Alg. (*) *p* < 0.001 vs. Method 1, ($) *p* < 0.001 vs. Method 2, (#) *p* < 0.001 vs. Method 3, (&) *p* < 0.001 vs. Method 4. (**E**) Porosity percentage (%) for the four scaffold preparation methods and the scaffolds functionalized with AuNp. (**F**) Degradation degree for the four scaffold preparation methods and (**G**) the functionalization of the four methods with AuNp. (**H**) Graphical representation of cell viability analysis for the four preparation methods functionalized with AuNp and AuNp+Alg. (*) *p* < 0.001 vs. without Np. Data are presented as the mean ± SE.

**Figure 3 polymers-14-03233-f003:**
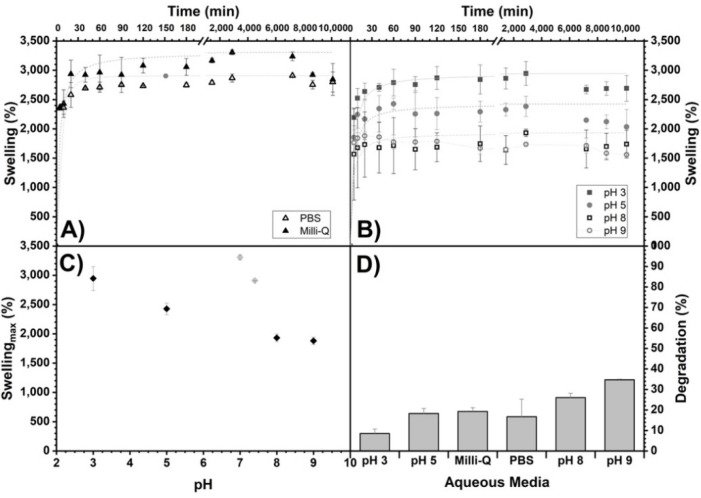
Swelling of alginate/chitosan scaffolds (Method 3) as a function of time in different aqueous media: (**A**) Milli-Q and PBS and (**B**) buffers prepared with Na_2_HPO_4_/citric acid for pH 3, 5, 8 and 9. In addition, their maximum swelling (*S*_max_) as a function of pH (**C**) and the percentages reached with respect to *S*_max_ at 5, 40 and 60 min are shown (see Table 1). Maximum swelling trend lines are shown as an eye guide. (**D**) Percentage of degradation (%) of the scaffolds of Method 3 (more crosslinking time) as a function of the aqueous medium exposed for 7 days.

**Figure 4 polymers-14-03233-f004:**
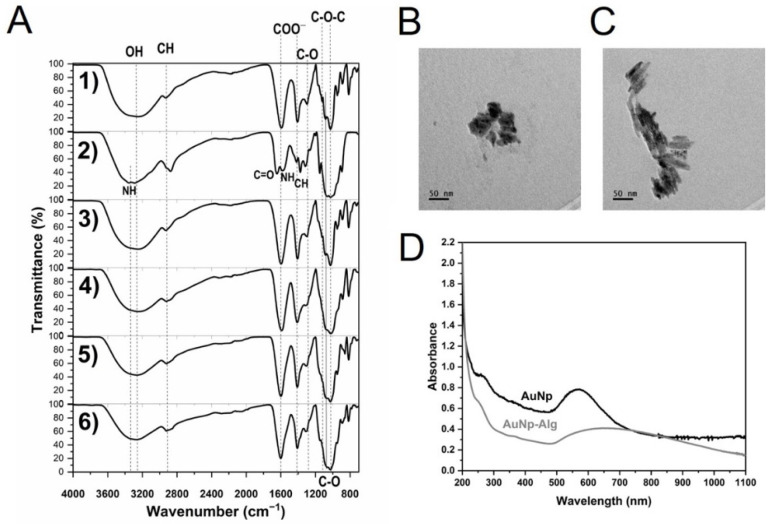
(**A**) FTIR-ATR spectra of: sodium alginate (1), chitosan (2) and alginate/chitosan scaffolds before (3) and after crosslinking (4, Method 3) and with incorporation of gold (5) or gold plus sodium alginate (6) nanoparticles. Vertical dotted and continuous lines are presented as eye guides. Representative electromicrographs of (**B**) AuNp and (**C**) AuNp+Alg obtained by TEM. (**D**) Gold nanoparticles plasmon.

**Figure 5 polymers-14-03233-f005:**
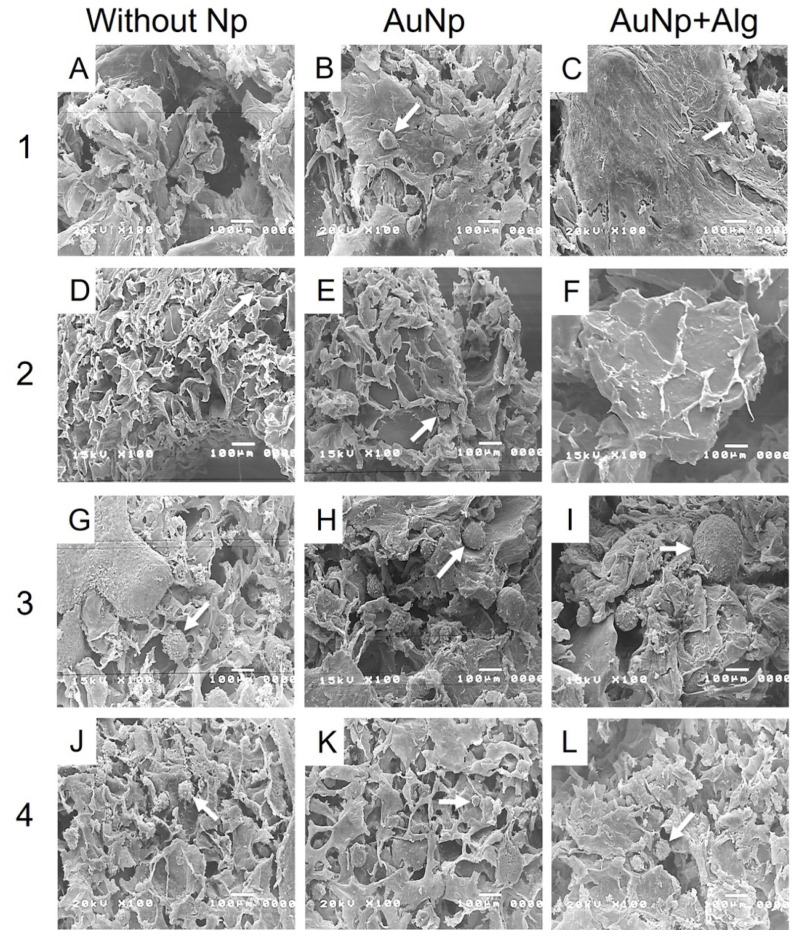
Representative electromicrographs of cross sections of the constructs. Method 1: (**A**) without Np, (**B**) AuNp and (**C**) AuNp+Alg. Method 2: (**D**) without Np, (**E**) AuNp and (**F**) AuNp+Alg. Method 3: (**G**) without Np, (**H**) AuNp and (**I**) AuNp+Alg. Method 4: (**J**) without Np, (**K**) AuNp and (**L**) AuNp+Alg. Identifier: arrow (spheroids).

**Figure 6 polymers-14-03233-f006:**
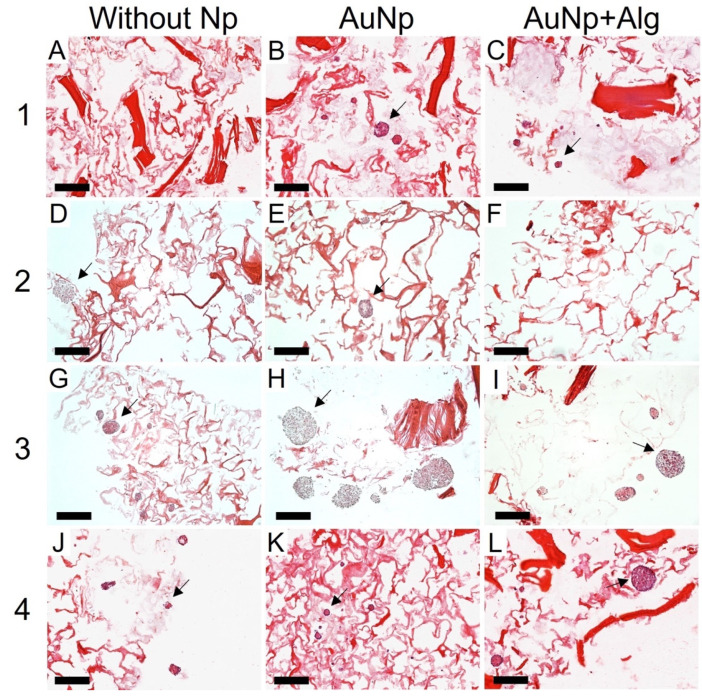
Representative photomicrographs of cross sections of the constructs. Method 1: (**A**) without Np, (**B**) AuNp and (**C**) AuNp+Alg. Method 2: (**D**) without Np, (**E**) AuNp, and (**F**) AuNp+Alg. Method 3: (**G**) without Np, (**H**) AuNp and (**I**) AuNp+Alg. Method 4: (**J**) without Np, (**K**) AuNp and (**L**) AuNp+Alg. H&E. Identifier: arrow (spheroids). Scale bar: 200 µm, original magnification: ×100.

**Figure 7 polymers-14-03233-f007:**
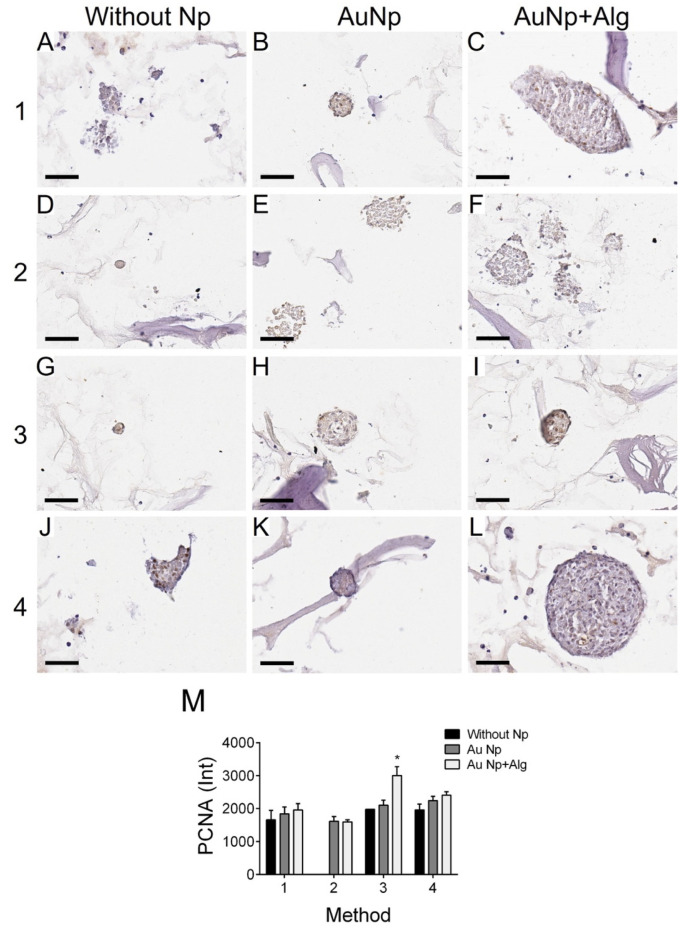
Representative photomicrographs of PCNA expression in different constructs. Method 1: (**A**) without Np, (**B**) AuNp and (**C**) AuNp+Alg. Method 2: (**D**) without Np, (**E**) AuNp and (**F**) AuNp+Alg. Method 3: (**G**) without Np, (**H**) AuNp and (**I**) AuNp+Alg. Method 4: (**J**) without Np, (**K**) AuNp and (**L**) AuNp+Alg. Scale bar: 50 µm, original magnification: ×400. (**M**) Graphical representation of the quantitative analysis of PCNA expression (Int) in the different constructs. (*) *p* < 0.001. Data are presented as the mean ± SE.

**Figure 8 polymers-14-03233-f008:**
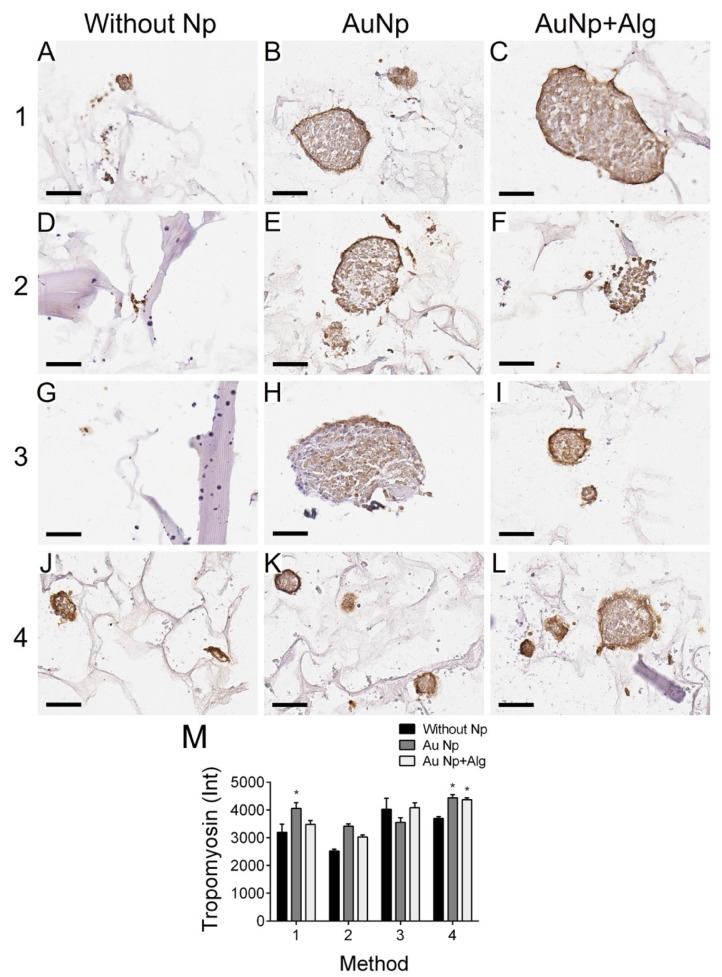
Representative photomicrographs of tropomyosin expression in different constructs. Method 1: (**A**) without Np, (**B**) AuNp and (**C**) AuNp+Alg. Method 2: (**D**) without Np, (**E**) AuNp and (**F**) AuNp+Alg. Method 3: (**G**) without Np, (**H**) AuNp and (**I**) AuNp+Alg. Method 4: (**J**) without Np, (**K**) AuNp and (**L**) AuNp+Alg. Scale bar: 50 µm, original magnification: ×400. (**M**) Graphical representation of the quantitative analysis of tropomyosin expression (Int) in the different constructs. (*) *p* < 0.001. Data are presented as the mean ± SE.

**Figure 9 polymers-14-03233-f009:**
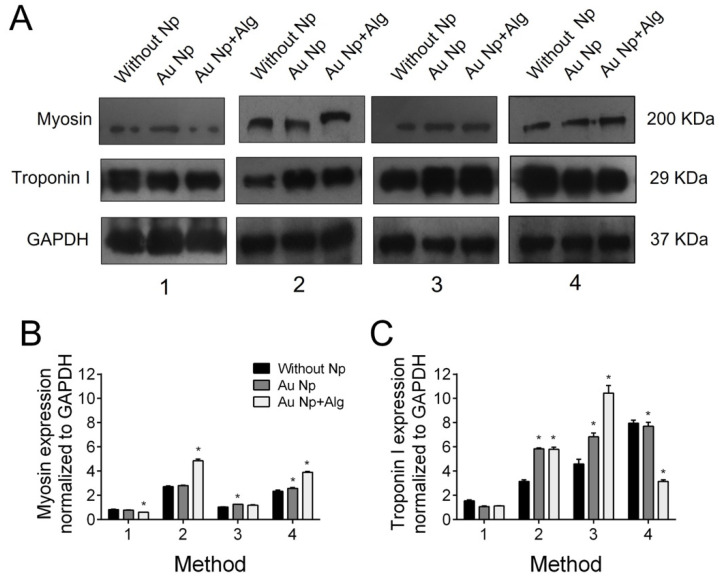
(**A**) Representative Western blot figure of myosin and troponin I in the different constructs: without AuNp and with AuNp and AuNp+Alg. Graphical representation of the quantitative analysis of (**B**) myosin and (**C**) troponin I expression. (*) *p* < 0.001. Data are presented as mean ± SE.

**Table 1 polymers-14-03233-t001:** Maximum swelling (*S*_max_) as a function of pH.

	Swelling Referred to *S*_max_ (%)		
Medium	5 min	40 min	60 min
pH 3	75	92	95
pH 5	76	97	100
Milli-Q	71	88	90
PBS	81	93	93
pH 8	53	57	58
pH 9	94	99	95

## Data Availability

The data presented in this study are contained within the article or are available as Appendix A.

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
