# Peer review of "Sodium Alginate/Chitosan Scaffolds for Cardiac Tissue Engineering: The Influence of Its Three-Dimensional Material Preparation and the Use of Gold Nanoparticles"

_polymers, 2022, doi:10.3390/polym14163233_

Round 1

Reviewer 1 Report

The work entitled "Sodium Alginate/Chitosan Scaffolds for Cardiac Tissue Engineering: The Influence of Its Three-Dimensional Material Preparation and the Use of Gold Nanoparticles" correlates with the scope and objectives of the journal "Polymers". The authors performed an interesting methodological analysis of the formulation of chitosan/alginate scaffolds in combination with gold nanoparticles. According to this reviewer's criteria, the work can be published after the following modifications and clarifications:

-It would be necessary to add a figure outlining the different methods of formulating the scaffolds. This would help to better understand the data presented.

-Based on the results presented it is not possible to understand the nature of the nanoparticles developed. It would be necessary to add uv-vis spectroscopy results in order to visualize the plasmon of the gold nanoparticles. In the TEM images (Figures 3 g and h) it would appear that the gold would be forming spheres that are included inside the polymer. It would be important to perform an EDS analysis to understand the distribution of the elements within the nanosystems.

-Although the importance of adding gold nanoparticles in the scaffolds has been demonstrated, the distribution of these nanosystems within the scaffolds remains to be determined. It may happen that when the NPs are included in the polymer solutions, they aggregate and distribute heterogeneously.

-The authors mention the importance of gold nanoparticles with respect to conductivity, however, they do not perform studies to verify if this property improves when adding the nanosystems.

-In the introduction it would be important to highlight the polyelectrolyte characteristics of the materials and their properties based on pH.

-The permeability determination method was developed based on the equation used, but the methodology used is not clear.

Reviewer 2 Report

Lines 39-40. “using calcium gluconate as a crosslinking agent” Could the authors please explain how calcium gluconate can act as a crosslinking agent? I always assumed that crosslinking implies the formation of covalent bonds between different parts of the molecule.

Line 102. “…and subsequently freeze-dried for 8 h at −49°C…”I would like to clarify the drying conditions. Perhaps you meant the trap temperature in a lyophilic dryer? What was the temperature on the shelf where the experimental material was dried? Such a low drying temperature, especially in just 8 hours, is surprising.

Line 102-103. “a vacuum pressure of 0.100 mBar.” I suggest writing “a pressure of 0.100 mBar.”, since a vacuum has no pressure.

Lines 98-122. It is necessary to adequately describe the preparation of composites. Chitosan does not dissolve in water, but from lines 98-99 it directly follows that it was dissolved in water, acetic acid was added later. What was the exact amount of alginate, chitosan and acetic acid used in the experiment? Lines 107-108. Exactly how much calcium gluconate was used and in what way? How much water was used for washing and how many times was this done? How was the quality of the washing controlled? How exactly, at what power, on what apparatus was sonication performed?

Lines 125-144. A detailed description of the synthesis is needed. What volumes of solutions were mixed? Under what conditions and at what volumes was centrifugation performed?

Line 218. “…and a vacuum of 0.09 mBar” I suggest replacing it with “…and a pressure of 0.09 mBar”

Lines 261-262. “Scaffolds and constructs were fixed with glutaraldehyde (4%) and dehydrated 261 through a series of graded ethanol concentrations (50° to absolute).” What was the purpose of cross-linking the materials with glutaric aldehyde before drying? Chemical reactions of the formation of Schiff bases with the amino groups of chitosan resulted in a different material that was not investigated by other methods in this work.

Line 280. “(Vector Laboratories)” You must specify the city and country of manufacture of the kit.

Lines 479-480. ” in this work the  use of calcium gluconate as a crosslinking agent is proposed, favoring cell adhesion and  cardiomyocyte growth.”

I cannot agree that the calcium gluconate used in this work acted as a crosslinking reagent. Even if we conclude that the formation of ionic bonds acts as a "crosslinking" factor, we must consider that there is only one carboxylic group in a single gluconic acid molecule. Thus, gluconic acid makes a salt with the amino group of chitosan and this is the end of all chemical interactions. No chemical bonds are formed between the gluconic acid residue and other polymers and particles in the scaffold. This is also confirmed by the fact that there is no significant difference between the spectra C and D in the IR spectra.

Round 2

Reviewer 1 Report

The authors complied with all requested modifications and suggestions. According to this reviewer's criteria, the paper can now be published.